# Design of large-span stick-slip freely switchable hydrogels via dynamic multiscale contact synergy

Zhizhi Zhang[1,2], Chenxi Qin[1,2], Haiyan Feng[1,2], Yangyang Xiang[1], Bo Yu[1], Xiaowei Pei[1], Yanfei Ma [1] ✉ & Feng Zhou [1] ✉

Solid matter that can rapidly and reversibly switch between adhesive and non-adhesive states is desired in many technological domains including climbing robotics, actuators, wound dressings, and bioelectronics due to the ability for on-demand attachment and detachment. For most types of smart adhesive materials, however, reversible switching occurs only at narrow scales (nanoscale or microscale), which limits the realization of interchangeable surfaces with distinct adhesive states. Here, we report the design of a switchable adhesive hydrogel via dynamic multiscale contact synergy, termed as DMCS-hydrogel. The hydrogel rapidly switches between slippery (friction ~0.04 N/cm²) and sticky (adhesion ~3 N/cm²) states in the solid-solid contact process, exhibits large span, is switchable and dynamic, and features rapid adhesive switching. The design strategy of this material has wide applications ranging from programmable adhesive materials to intelligent devices.

As a novel potential adhesive material, switchable adhesives have been widely researched in climbing robotics[1], actuators[2], wound dressings[3], bioelectronics[4], etc. Their main feature is the ability to achieve on-demand attachment and detachment. A key challenge for switchable adhesive materials in these application fields, is the realization of large-span tailored adhesion, which involves ultra-low adhesion and lubrication during detachment switching. The slippery state endows the switchable adhesive with the advantage of easy detachment without interface residue, otherwise the contact interface would be polluted[5], which is not favorable for many sophisticated engineering applications (e.g., climbing robots, medical instruments). Scientists have attempted to address this issue primarily by considering cohesion and interfacial adhesion. Large-span switchable adhesive materials mediated by cohesion, generally rely on switchable mechanical properties to regulate their adhesion strength[6-9]. However, using this reversible framework, most smart adhesive materials suffer from defects such as the residue left on the contact surface after detachment, slow regulation between adhesive and non-adhesive state, highly complex control system, or high adhesion during the detachment. Recently, interface-based smart adhesive materials have achieved long-term adhesion stability through the chemical covalent crosslinking of the interface or the entanglement between contact surfaces[3,10]. Generally speaking, strong covalent bonds involve strong adhesion to the interface, which is usually accompanied by long separation time and requires strong external energy[7].

Non-covalent bond interface adhesion through van der Waals forces, hydrogen bonds, coordination or cation-π interactions[11], have more potential for achieving residue-free, rapidly adjustable, switchable adhesive materials due to their reversible interaction nature with surfaces, wherein the areal contact is highly crucial for non-covalent interface adhesion[12] since it largely determines the interaction strength. Over the past decades, major progress has been made on switchable adhesive materials that rely on interface contact to control the adhesion strength at micro-scale, meso-scale, and even macro-scale. For the design of switchable adhesive materials at the micro-scale, a representative example involves mussel-inspired biomimetic method, which is normally based on reversible screening of the catechol groups at the molecular level[13-15]. However, the change of contact

[1]State Key Laboratory of Solid Lubrication, Lanzhou Institute of Chemical Physics, Chinese Academy of Sciences, 730000 Lanzhou, China. [2]College of Materials Science and Opto-Electronic Technology, University of Chinese Academy of Sciences, 100049 Beijing, China. ✉e-mail: mayanfei@licp.cas.cn; zhouf@licp.cas.cn

area at the molecular level cannot be guaranteed without contact at other scales, thus uncontrollable/chaotic contacts at other scales may weaken the switching of adhesiveness, leaving a substantial portion of multiscale contact underutilized in contact switching during adhesion regulation. Incorporating mussel-mimetic smart polymers into micro/nano-structures could increase the effective contact area at the mesoscopic scale and improve high-low adhesion strength at the same time[16–18]. Unfortunately, structure-assisted smart adhesives essentially only rely on adhesion regulation at the molecular scale. Moreover, the low-adhesion threshold of most smart adhesive materials exceeds 10 kPa[7,19–21], which is sufficient to lift a 2.5 kg weight with a contact area of $5 \times 5$ cm. Thus, realizing large-span transition between ultra-low adhesion (slippery) and high adhesion (sticky) states during attachment/detachment is difficult to attain for solid adhesion at a simple and single scale. This necessitates a synergy of dynamic multiscale contact to achieve interchangeable solid surfaces with distinct performances.

In nature, the characteristic of dynamic multiscale contact for adhesion are widespread in organisms such as lizard tail autotomy[22] and gecko toe adhesion[23,24]. This multiscale interface strategy endows these creatures with superpowers to run freely on horizontal, sloping, vertical, and even inverted surfaces, or quickly and easily shake off their tails to distract the enemy. By imitating biological characteristics, the development of bio-inspired adhesives integrating the functions of high and low adhesion based on a single narrow scale contact, has attained many important achievements[25]. In fact, one of key factors in achieving higher adhesion for adhesive materials is the existence of a large number of activated adhesive groups on the surface at a microscopic scale, and the precondition for these adhesive groups to function is that the two surfaces have a sufficiently close distance, i.e., sufficient contact at mesoscopic scale (Fig. 1b). Conversely, lower adhesion is often associated with deactivated adhesive groups and insufficient contact. Fundamentally, achieving large-span switch between high and low adhesion is synergistic modulation of dynamic multiscale contact (Fig. 1c)[26]. However, it is an immense challenge to dynamically modulate contacts at each scale and synergize the dynamic multiscale contacts.

Herein, we propose the synthesis of a switchable adhesive hydrogel by combining dynamic multiscale contact with coordinate regulation. Through this regulation, the smart hydrogel with dynamic multiscale contact synergy (denoted as DMCS-hydrogel) exhibits a large-span, is switchable and dynamic, and features rapid adhesion switching between the sticky and the slippery states, wherein ultra-low adhesion and high-roughness endow DMCS-hydrogel with low friction force at high temperature[27]. The adhesive hydrogel can be applied to various substrates and displays superior adhesion properties. The characteristics of fast switching, no residue, and high fidelity significantly extend the applicability of the smart hydrogel in tissue replacements, surgical implants, surgical instruments, soft-robotics, wearable electronics, and coatings.

## Results

### Realization of dynamic multiscale contact synergy (DMCS)
As a proof-of-concept, we used poly(dopamine methacrylamide) (PDMA) as the adhesive component[17] (Supplementary Fig. 1, $^{1}$H NMR and $^{13}$C NMR of DMA), poly(N-isopropyl acrylamide) as the smart component, poly(acrylamide) (PAAm) as the elastic component[28], and poly(acrylic acid) (PAAc) as the lubricating component in our copolymer p(AAm-co-AAc-co-NIPAAm-co-DMA) as an exemplary DMCS-hydrogel. Acrylamide (AAm), acrylic acid (AAc), dopamine methacrylamide (DMA), and N-isopropyl acrylamide (NIPAAm) monomers were mixed in different proportions and covalently crosslinked with UV light. p(AAm-co-AAc-co-NIPAAm) and p(AAm-co-AAc-co-DMA) hydrogels were used as control samples. Our adhesion switch is mainly contributed by two parts: molecular-scale

(microscopic) regulation of mussel-adhesive molecules and mesoscale regulation based on modulus and roughness (Fig. 1c). Currently, catechol-functionalized adhesives inspired by mussels have been widely used in dry, wet, and under water conditions, whose adhesion to the surface of substrates originate from abundant weak interactions, including van der Waals forces, hydrogen bonds, coordination or cation-π interactions, and so on[11]. Reversible screening of the catechol groups can dynamically and efficiently switch the interface adhesion[29]. As shown in Fig. 1a, the adhesion switching of DMCS-hydrogel at the molecular scale depends on the working and non-working of mussel-inspired adhesive molecules as a result of water in or water out on the hydrogel surface. Specifically, the DMCS-hydrogel consists of long carbon chains, catechol groups, carboxyl groups, and N-isopropyl groups. Below the lower critical solution temperature (LCST) of DMCS-hydrogel, the N-isopropyl groups can easily form intermolecular hydrogen bonding with adjacent water molecules, thus binding a lot of water molecules in the hydrogel. The surface of the pristine DMCS-hydrogel was filled with air, which preferentially exposed the non-polar parts of the backbone and catechol groups toward the air to minimize surface energy, thereby bringing molecular-scale adhesion to the hydrogel surface. The adhesion property can be annihilated in a high-temperature environment. Above the LCST, the intramolecular hydrogen bonds in DMCS-hydrogel were disrupted, resulting in a large amount of water molecules spilling onto the surface of the gel. The surface water determines the reorientation of the hydrophobic and hydrophilic groups, resulting in the migration of carboxyl groups to the surface, to minimize interfacial free energy. This process effectively shields the adhesive catechol groups from interacting with the substrate surface and increases the lubricating effect[30]. Figure 1d further illustrates the contribution of catechol groups to interface adhesion at the microscopic scale—the adhesion strength gradually increases with the increase of the adhesive component poly (dopamine methacrylamide) (PDMA). The adhesion performance of the hydrogels without PDMA may be related to some weak interactions based on other functional groups (e.g., van der Waals forces originating from carbon-carbon backbone)[31], which are easily annihilated by water molecules[32]. When the hydrogels have the same interface adhesion at the microscopic scale, the adhesion strength decreases with the increasing modulus due to the mesoscopic contact changes (Fig. 1e); the change in modulus was obtained by slightly changing the crosslinking agent (Supplementary Fig. 4 and Fig. 5). For DMCS-hydrogel, reversible non-covalent crosslinking (conformational transition) based on the N-isopropyl groups not only induces concurrent changes in the modulus and surface roughness at the mesoscopic scale (Supplementary Fig. 6, Fig. 7 and Fig. 8), but also induces reversible screening of adhesive molecules at the microscopic scale. Furthermore, micro-scale and meso-scale contacts are concurrently controlled by the same variable, i.e., temperature, and the two act synergistically, which is crucial for achieving large-span, reversible, dynamic, and rapid adhesion switching between the slippery and sticky states. As a result, the adhesion strength of DMCS-hydrogel exhibits distinct differences between high and low temperature, and the gap increases with the increase of N-isopropyl group concentration in the hydrogel. By contrast, the p(AAm-co-AAc-co-DMA) hydrogel (without smart N-isopropyl group) exhibits negligible adhesion changes in response to temperature (Fig. 1f). In a typical case, DMCS-hydrogel sample ($20 \times 20 \times 2$ mm) showed high adhesion (stick) to the iron sheet at low temperature and did not debond even when the hydrogel was elongated. However, after heating the iron sheet with a silicone rubber heating piece, the hydrogel immediately detached and even slid on the surface of the steel sheet (Fig. 1g). Moreover, DMCS-hydrogel, exhibits a shear modulus of 36.33 kPa and the ability to stretch to more than 7 times its original length, and exhibits excellent mechanical properties (Fig. 1h, i).

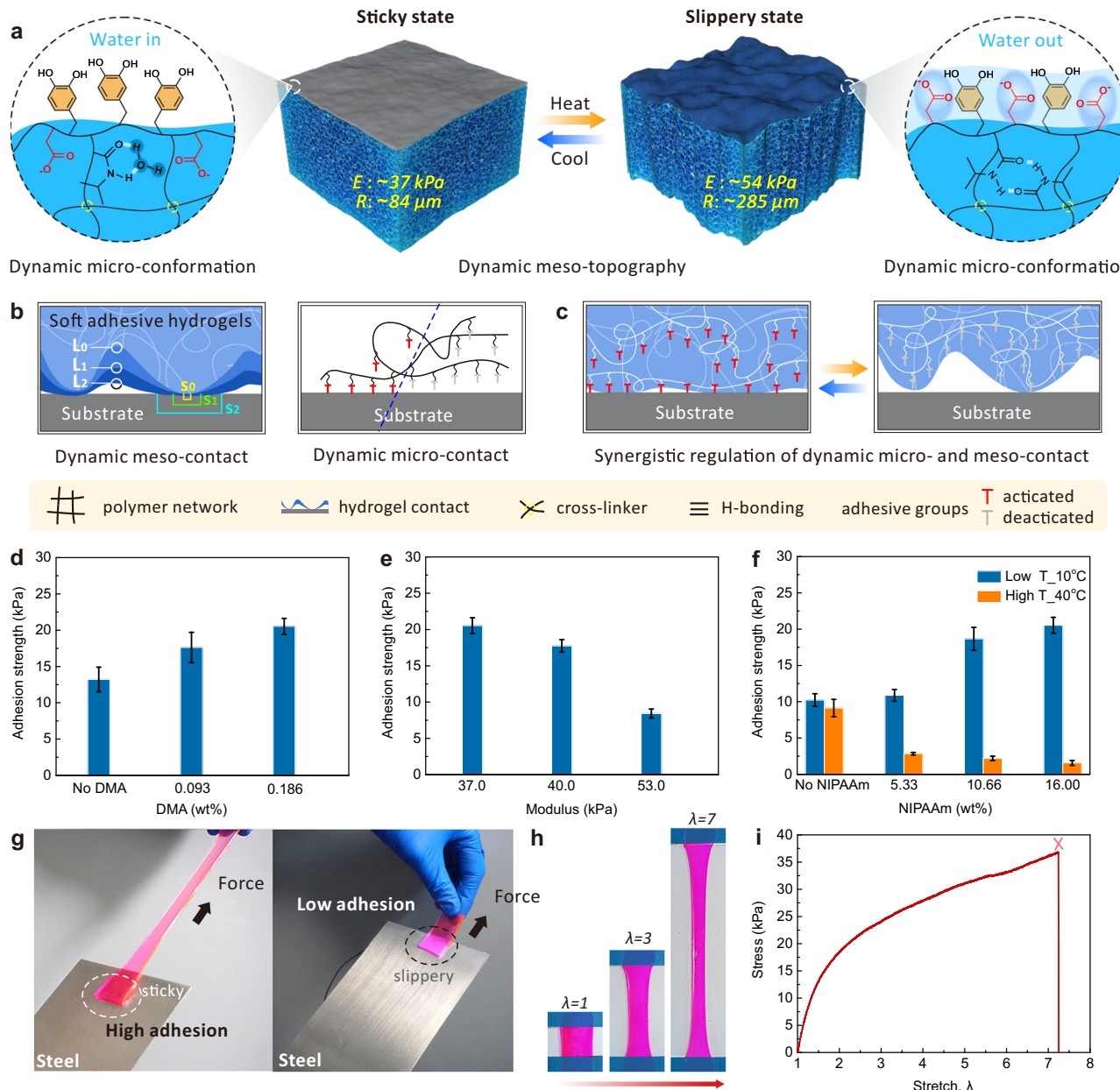

**Fig. 1 | Design and realization of DMCS-hydrogel. a** The sticky and slippery switching mechanism of DMCS-hydrogel via dynamic multiscale contact synergy originating from the dynamic meso-topography and micro-conformation. E: elastic modulus, R: roughness. **b** Schematic diagram of meso-scale contact and micro-scale contact. Left, dynamic meso-contact via roughness and modulus change of soft adhesive materials. Right, dynamic micro-contact based on the activation and deactivation of adhesive groups. $L_0$, $L_1$, $L_2$: distance between hydrogel surface and substrate; $S_0$, $S_1$, $S_2$: contact area between hydrogel surface and substrate. **c** The

synergistic regulation of dynamic micro- and meso-scale contact. The effect of adhesive components (PDMA) (**d**), material modulus (**e**), and smart components (PNIPAAm) (**f**) on adhesion strength. **g** The sticky and slippery state of the DMCS-hydrogels against iron sheet at high and low temperatures. Photographs (**h**) and nominal stress versus stretch curve (**i**) of the DMCS-hydrogel, stretched to more than 7 times its original length. The hydrogel was colored with Rhodamine B for visualization. Error bars represent the standard deviation from at least three replicates. Data in **d**, **e**, and **f** are presented as mean values ± SD. λ stretch.

## Dynamic multiscale contact mechanism

To validate this design concept, we first observed the evolution of the contact formation between round glass (Φ = 5.4 cm, H = 20 mm) and DMCS-hydrogel via a home-made setup, as shown in Fig. 2a. In brief, the DMCS-hydrogel cube was placed on the temperature controller, attached to the load cell, and rapidly approached the quartz glass with a diameter of 2.7 cm at a rate of 40 cm s$^{-1}$, until the 0.4 kPa designed value. The contact images were captured using a digital camera. The snapshots of the contact images show a gradual contact process from 0, 0.1, 0.4, to 4 s, wherein the dark region (dark blue) is in contact with the glass, and bright region (light blue) is the untouched part. At the

low temperature, the contact started locally, and then gradually and irregularly developed across the whole region within 4 s, while the contact process took longer at high temperature (Fig. 2b, Supplementary Movie 1). Specifically, the effective contact area of the DMCS-hydrogel at low temperature accounted for about 80% at 0.4 s, while there was almost no contact at high temperature. Until about 40 s, the high-temperature hydrogel was completely in contact with the quartz glass. The difference in contact rates between the two was almost 100 times. In addition, for high temperature, the effective contact area at contact equilibrium was still smaller than that at low temperature (Fig. 2c). The experimental results demonstrated that the high- and

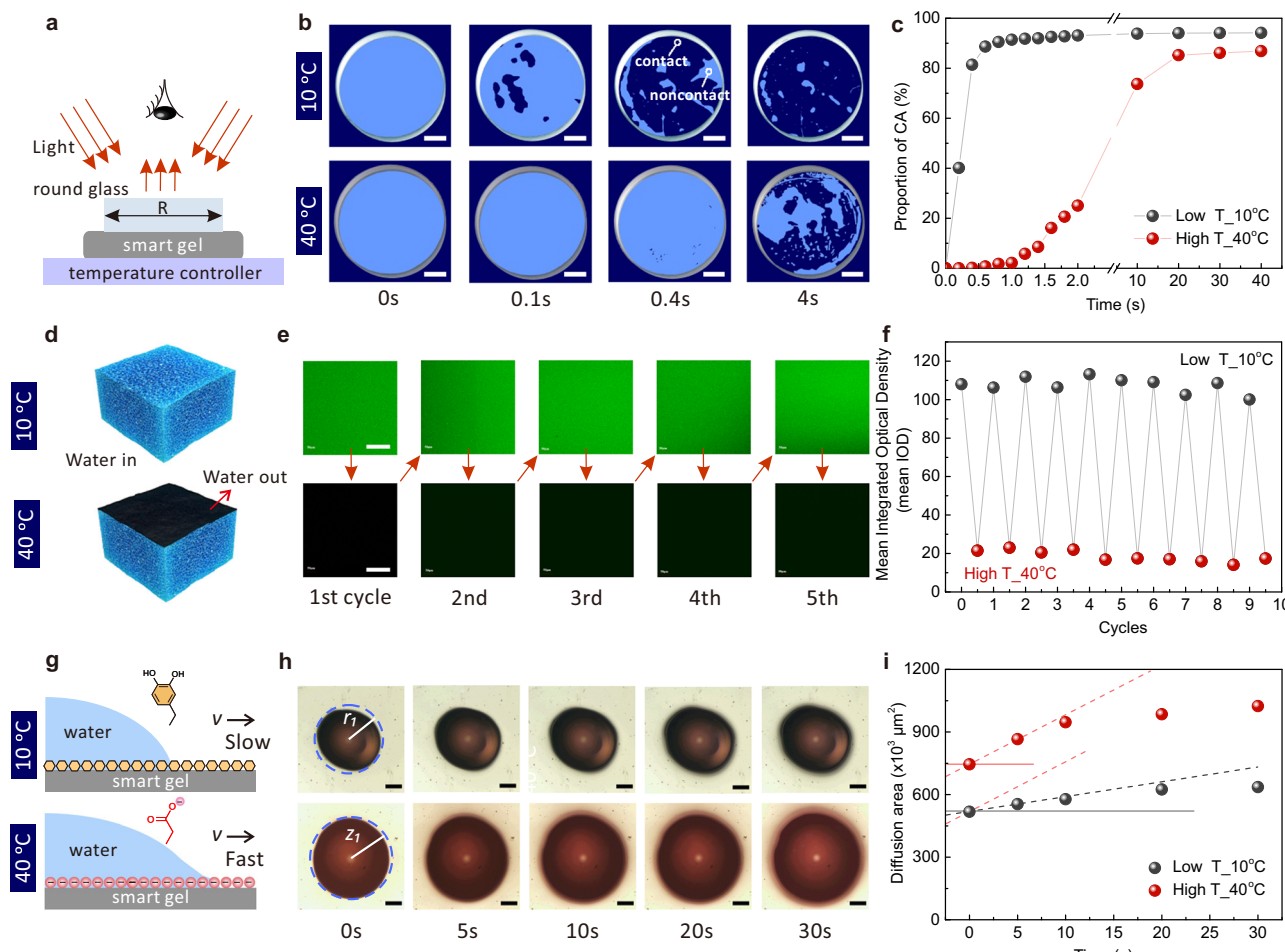

**Fig. 2 | Dynamic multiscale contact mechanism of DMCS-hydrogel. a** A schematic diagram to observe the contact evolution between DMCS-hydrogel and quartz glass. **b** Snapshots of the contact images of the low- and high-temperature DMCS-hydrogel. The bright region was in contact with the glass and the dark region was not in contact. Scale bar, 5 mm. **c** The relationship between the contact time and the percentage of effective contact area. **d** Surface wetting and dewetting of DMCS-hydrogel were observed by a confocal microscope. **e** The fluorescent hydrogel network was covered by water at high temperature and recovered at low temperature. This process can be repeated. Scale bar, 100 μm. **f** The mean IOD showed the wetting and dewetting cycles of the DMCS-hydrogels under high and low temperature conditions. **g** Schematic diagrams of the dynamic wetting of fluorescent water droplets on the surface of DMCS-hydrogel. **h** Snapshots of the dynamic wetting images of low- and high-temperature hydrogel. Scale bar, 200 μm. **i** The corresponding curve of time versus diffusion area.

low-temperature DMCS-hydrogels had a large difference in mesoscopic-scale contact, including the contact rate and the true contact area (Supplementary Fig. 11 and Fig. 12), due to the temperature-induced synergy of roughness and modulus. Furthermore, we further investigated the independent effects of the interfacial roughness and modulus on the contact process of soft materials via single variable. As shown in Supplementary Fig. 9, during the gradual contact of the soft matter with the substrate, the low-modulus and high-modulus hydrogels had the same initial contact area (0.1 s contact), but the low-modulus hydrogel had a faster contact rate. This means that there is a strong correlation between modulus and contact rate. In contrast, the initial contact area of low-roughness hydrogel and high-roughness hydrogel were quite different, but the contact rate was similar, indicating that roughness determined initial contact area (Supplementary Fig. 10). In the dynamic multiscale contact system, temperature not only induced changes in contacts at the mesoscopic scale, but also affected the contacts of adhesive molecules at the microscopic scale, which is attributable to the reversible water out and water in. As shown in Fig. 2d, the hydrogel network with fluorescence was exposed to air at low temperature, hence large amounts of fluorescent signals were detected by the confocal laser scanning microscope. By contrast, the fluorescence signal of the polymer vanished at

high temperature because the hydrogel surface was wetted by water molecules. In particular, the water in and water out of DMCS-hydrogel can be recycled and remained stable after 10 cycles (Fig. 2e, f, Supplementary Figs. 13 and 14). The dynamic wetting of DMCS-hydrogel led to the reorientation of the hydrophobic and hydrophilic groups at the outermost interface between 5 and 10 Å, resulting in reversible screening of the adhesive (catechol) groups through the conformation transition of carboxyl groups (Fig. 1a)[13,16,33]. Using the dynamic wettability of water droplets on the surface of DMCS-hydrogel, we further verified the reorientation of hydrophobic (long carbon chains, catechol) and hydrophilic (carboxyl) groups on the surface of DMCS-hydrogel at alternating temperatures (Fig. 2g). A 4 μL dyeing water droplet was placed on the low- and high-temperature DMCS-hydrogels' surface. The process of dynamic wetting in the vertical and horizontal directions was recorded using microscopes, with the wetting radius $r_1$ at low temperature and $z_1$ at high temperature. As shown in Fig. 2h and Supplementary Movie 2, the water droplets spread faster on the high-temperature hydrogel than on the low-temperature hydrogel from the top view and the corresponding contact angles of high temperature decrease faster than that of the low temperature from the side view (Supplementary Fig. 15 and Fig. 16). Specifically, the diffusion rate at high temperature ($23.4 \times 10^3$ μm²/s) was 3 times higher

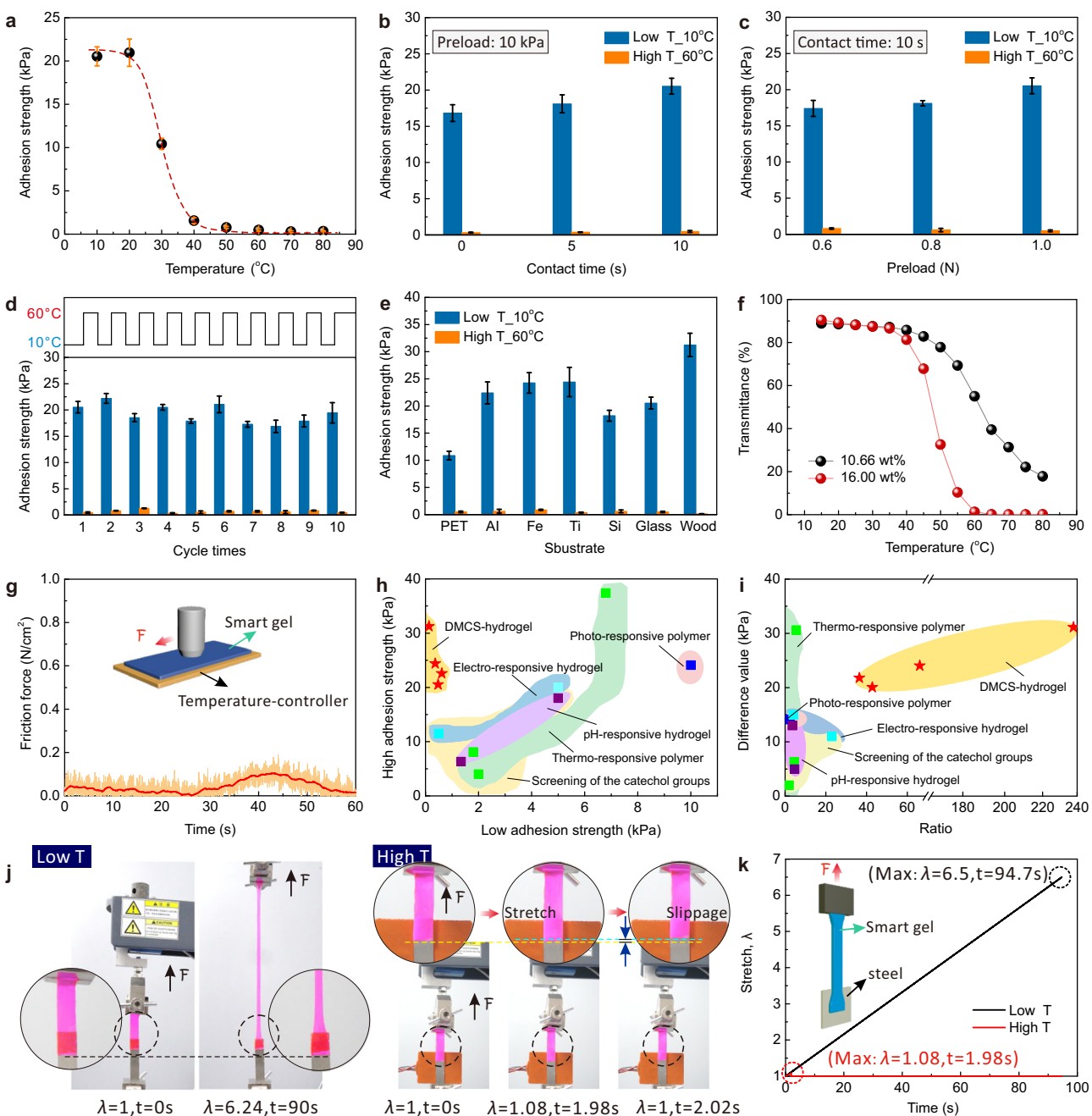

**Fig. 3 | Switchable adhesion properties of hydrogels via dynamic multiscale contact synergy. a** The relationship between the temperature and adhesion strength of DMCS-hydrogel. The effects of contact time (**b**) and preload **c** on adhesion strength at high and low temperatures against a glass probe. **d** Thermo-responsive switchable adhesion cycles of DMCS-hydrogel at high and low temperature. **e** The switchable adhesion properties of the smart hydrogel can be applied to a variety of solid substrates including inorganic and organic surfaces. **f** The variation of LCST in DMCS-hydrogel with different pNIPAAm content. Transmittance vs. temperature captured at 400 nm wavelength by UV-vis spectrometer. **g** Friction curves in time domain at high temperature. Ultra-low adhesion and high roughness enable the DMCS-hydrogel to have low friction force at high temperature. **h**, **i** Comparison of adhesion properties between DMCS-hydrogel, other reported switchable adhesive hydrogels, and other smart adhesive materials[1,4,13,14,16,19,35,36]. The data used are summarized in Supplementary Fig. 30 and Supplementary Table 2. **j** Shear adhesion tests at low temperature and high temperature and **k** the corresponding stretch-time curve. The adhesion area was 1 cm². Error bars represent the standard deviation from at least three replicates. Some error bars in **a** are hidden by the symbols. Data in **a**, **b**, **c**, **d**, and **e** are presented as mean values ± SD. F force, λ stretch, t time.

than that at low temperature ($7.2 \times 10^3$ μm²/s), indicating that the high-temperature surface has more hydrophilic functional groups (Fig. 2i).

## Large-span switchable adhesion

As previously demonstrated, temperature controlled the dynamic multiscale contact of DMCS-hydrogel. To further investigate the effect of temperature, the adhesion properties of hydrogel to glass was

detected at different temperatures from 10 to 80 °C (Fig. 3a, Supplementary Fig. 17 and Fig. 18). Unless otherwise specified, all adhesion test conditions are 10 s contact time and 10 kPa preload. It is evident that the adhesion strength of DMCS-hydrogel decreased with the increasing temperature, and DMCS-hydrogel had a higher adhesion strength (~21 kPa) in the low-temperature region (from 10 to 20 °C). Then, the adhesion strength started to decrease after 20 °C and stabilized by

60 °C (~0.4 kPa). Theoretically, for interface adhesion, increasing contact time and preload can both improve the interface adhesion strength, probably due to the local conformational rearrangements and the sufficient contact of adhesive molecules, resulting in more effective contact 'bonds' with the substrate surface[34]. For this multiscale adhesion system, the adhesion strength of DMCS-hydrogel increased slightly with the increase of contact time (constant preload: 10 kPa) and preload (constant contact time: 10 s) at low temperature, and exhibited negligible change at high temperature, indicating that the rearrangement of the molecular conformation on the hydrogel surface and interfacial contact were completed in a very short time, which resulted in the fast switching of the adhesion with temperature alternation (Fig. 3b, c and Supplementary Fig. 29). To verify the rapid bonding and debonding properties of DMCS-hydrogel, it was attached to an iron sheet without preload and then detached in situ above the LCST (60 °C in this case). As a result, the hydrogel could easily lift the 100 g iron sheet at low temperature and then quickly release it after the temperature increased (Supplementary Fig. 28). Notably, when the temperature of DMCS-hydrogel decreased below LCST, the interface adhesion was reactivated, exhibiting a full reversible signature. As shown in Fig. 3d, with the cycling of the temperature, the adhesiveness of DMCS-hydrogel can be reversibly and repeatedly switched, spanning 10 cycles. Under a condition of simple rehydration to prevent water loss, the adhesion performance of DMCS-hydrogel remained stable during the 5-day adhesion test without a significant decline (Supplementary Fig. 21) and can be stably and reversibly switched about 50 times (Supplementary Figs. 19 and 20). Furthermore, the switchable adhesion was effective, and was suitable for wide-ranging solid substrates from organic to inorganic surfaces, including polyethylene terephthalate (PET), silicon wafer (Si), glass, iron (Fe), titanium (Ti), aluminum (Al), and wood, wherein the high adhesion (32 kPa, 3.2 N/cm²) of DMCS-hydrogel was 240 times higher than that of the low adhesion (0.13 kPa, 0.013 N/cm²) (Fig. 3e). The LCST of DMCS-hydrogel can be modulated without a marked alternation in the adhesive property, thus yielding smarter soft materials that satisfy application-specific requirements. As shown in Fig. 3f, the LCST can be varied between 45 and 60 °C by controlling the PNIPPAm content in the hydrogel. Here, we found that DMCS-hydrogel not only exhibits low adhesion at high temperature, but also low friction (Fig. 3g). As shown in the Supplementary Fig. 29 and Supplementary Movie 3, a weak horizontal force can promote the hydrogel with high adhesion at low temperature to slide on the surface of a high-temperature iron substrate. This was expected for the DMCS-hydrogel, as the ultra-low adhesion between the contact surface and high roughness at high temperature already have the necessary conditions for achieving low interfacial friction[27]. This was also demonstrated by the shear adhesion test of DMCS-hydrogel, by taking the location of gel and iron sheet as a baseline. The hydrogel could be stretched to about 6.5 times its original length without debonding at low temperature (The adhesion area was 1 cm²). When the adhesive region was heated with a silicone rubber heating sheet, the hydrogel rapidly separated from the iron sheet without deformation (Fig. 3j, Supplementary Fig. 22 and Supplementary Movie 4). Overall, DMCS-hydrogel demonstrated excellent adhesion switching between high adhesion (sticky) and ultra-low adhesion state (slippery), far exceeding the values seen in several reported hydrogels (Fig. 3h, Supplementary Table 2). The DMCS-hydrogel displayed large-span adhesive switching, and the adhesive switching value was more than ~240 times (Fig. 3i). Moreover, we incorporated photo-thermal $Fe_3O_4$ nanoparticles into the hydrogel (Supplementary Figs. 23, 24, and 25). The adhesion properties of hydrogels were managed by utilizing local and remotely controlled temperature responses (Supplementary Fig. 26).

## Application of DMCS-hydrogel
Due to the advantages of large-span and reversibility and dynamic and rapid adhesion switching without preloading, DMCS-hydrogel is the ideal material for climbing robotics. As shown in Fig. 4a, DMCS-hydrogel with dynamic adhesive switching based on adhesion and friction were installed on the track of a mobile device, to control its movement on a vertical iron plate, where $F_A$ and $F_F$ represent the adhesion and friction during the movement, respectively. At low temperature, the climbing robot can crawl on an almost vertical metal surface relying on adhesion. The moving process was nearly a uniform linear motion (Fig. 4b). However, under elevated temperature conditions, it fell down after crawling for a period of time. This failure process was mainly divided into three parts: (i) uniform motion, (ii) debonding process, and (iii) free fall (Fig. 4c). The reason for this result was that DMCS-hydrogel covered on the track was initially cold, exhibiting high adhesion. The track consistently rolled during the movement of the mobile device on the high-temperature iron substrate. Although the local hydrogel contacted with the substrate, quickly switched from high adhesion to low adhesion, the low-adhesion hydrogel was rapidly transferred to the rear part of the device with the rotation of the track. At the same time, most of the contact region was replaced by the low-temperature hydrogel with adhesiveness and maintained movement of the device. As the device moved, the adhesion strength of the track at the rear decreased, peeling off from the contact interface and eventually causing the failure of the adhesion (i→ii→iii). To further demonstrate the precise controllability of the motion behavior of mobile devices with DMCS-hydrogel tracks, we designed and fabricated an intelligent temperature-controlled substrate, which consisted of two-part temperature regulators to program the temperature of localized regions on the iron substrate. The result shows that the temperature in the hot and cold areas is constant and their boundary is clear (Supplementary Fig. 27). The mobile device moved with constant power from top to bottom on the vertical intelligent substrate, passing through regions of low temperature and high temperature in sequence (Fig. 4d). As a result, the climbing robot moved at a constant speed for 5 s at low temperature, then promptly finished the uniform motion (i), debonding process (ii), and free fall (iii) within 0.4 s after entering the high-temperature region (Fig. 4e and Supplementary Movie 5). The application of the DMCS-hydrogel is not limited to such simple devices. In the future, we hope to expand the application of the DMCS-hydrogel to other medical devices and intelligent robots.

## Discussion
In this study, we strategically constructed a dynamic multiscale adhesive system based on micro-scale and meso-scale contact synergies by using molecular conformational rearrangements, modulus modulation, and roughness changes. The corresponding DMCS-hydrogel exhibited unusual material behavior, which can be switched in a reversible, dynamic, and rapid manner between sticky and slippery solid states. We foresee that the exposure of the adhesive mechanism in the dynamic multiscale contact system could provide guidelines for the design of more soft adhesive materials with novel capabilities for advanced applications. Although this work was mainly focused on the NIPAAm-based hydrogel with wetting, modulus, and roughness change, our approach and mechanism may be extended to design other high-performance smart soft materials based on the coupling of mesoscopic and micro-scale contacts, thus widening the application range of smart materials.

## Methods
### Synthesis of dopamine methacrylamide (DMA)
Ten grams $Na_2B_4O_7 \cdot 10 H_2O$ and 4 g $NaHCO_3$ were dissolved in 100 mL deionized water and bubbled with $N_2$ gas for 30 min. Next, 5 g of DOPA-HCl was added into the reaction flask. After stirring for 15 min, 4.7 mL MA was dissolved in 25 mL THF, and then added to the above reaction solution drop by drop. The reaction mixture was stirred

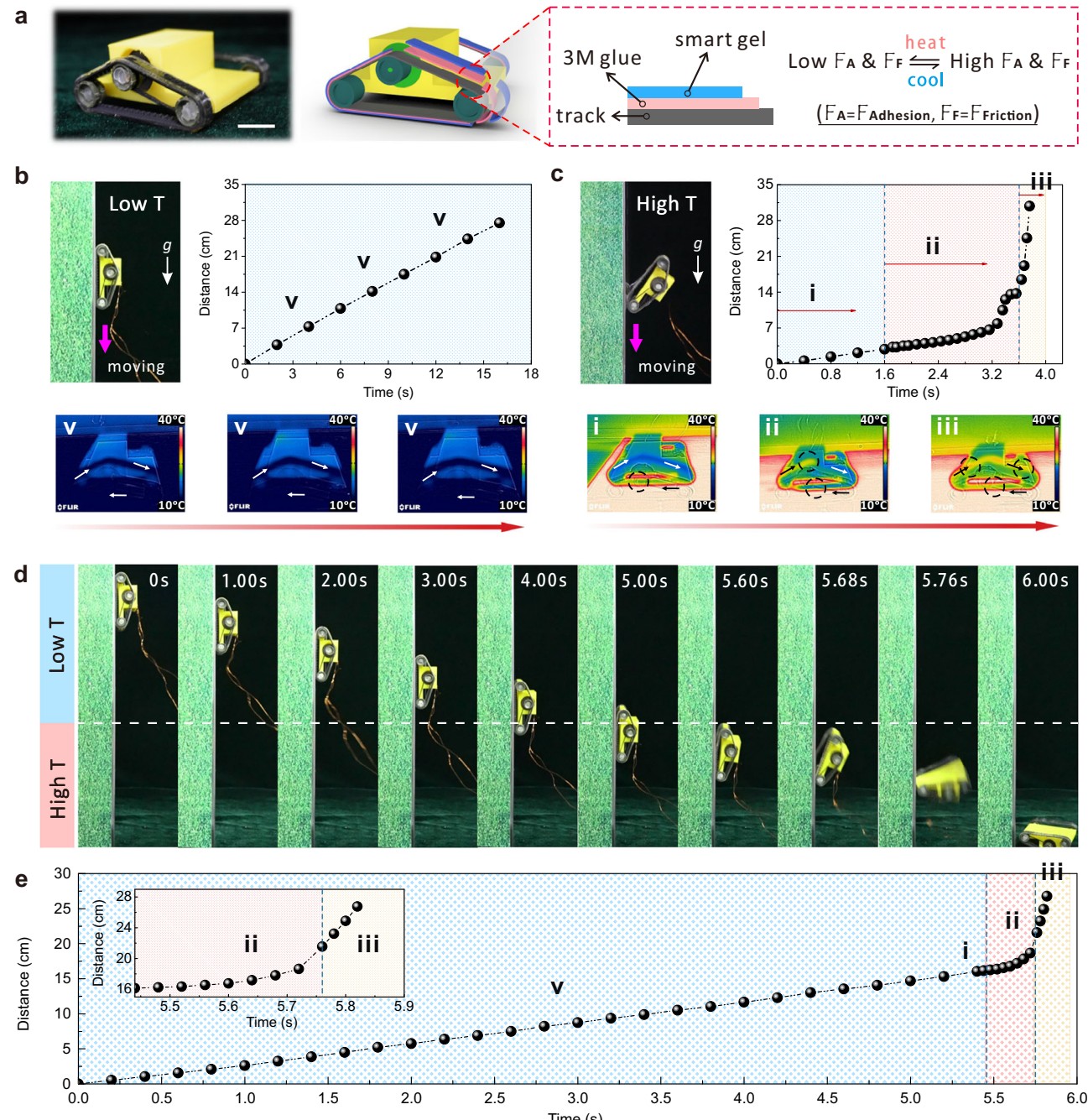

**Fig. 4 | Application of DMCS-hydrogel in a mobile device. a** The physical image of the mobile device manufactured using 3D printing technology. Its tracks were equipped with thermally responsive DMCS-hydrogels that caused a transition from high adhesion and high friction (High $F_A$ and $F_F$) to low adhesion and low friction (Low $F_A$ and $F_F$) during heating and cooling. Scale bar, 15 mm. Motion capture images of the mobile device crawling on a vertical metal plate at low temperature (**b**) and high temperature (**c**), as well as the related time-distance curve. Infrared thermal images showed the heat transfer process from the temperature-controlled substrate to DMCS-hydrogel. **d** The precise controllability of the motion behavior of mobile devices on the intelligent temperature-controlled substrate consisted of a two-part temperature regulator and **e** the associated time-distance curve. *v*: uniform linear motion at low temperature; i: motion curve during heating; ii: the process of stripping; iii: free fall. g: gravitational acceleration.

overnight under $N_2$ gas protection at room temperature. The pH of solution was kept above 8.0 by adding NaOH solution. After 24 h, the obtained solution was acidified with HCl (1 mol/L) until pH = 2. The solution was extracted with 100 mL of ethyl acetate for three times, the upper organic phase was taken. The water-soluble impurities in the organic phase were removed by NaCl solution, and then the solution was dried by anhydrous $MgSO_4$. Finally, the crude product was obtained after removing $MgSO_4$ and the solvent. The off-white product, DMA, was obtained by recrystallization from EtOAc. The

chemical structure of DMA was characterized by ¹H NMR (400 MHz) and ¹³C NMR analysis using a Bruker AVANCE III HD NMR spectrometer.

### Synthesis of the copolymer poly (DMA-PFOMA)
FOMA (2.3408, 5 mmol), DMA (0.11 g, 0.5 mmol), AIBN (0.016, 0.1 mmol) were dissolved in 2 mL DMF. At 80 °C, the solution was stirred for 12 h under the protection of $N_2$. The crude product was purified with n-hexane, and then a white solid powder was obtained. The poly (DMA-PFOMA) can be dissolved by 1,1,1-Trichlorotrifluoroethane. A

hydrophobic surface can be obtained by coating the solution on the surface of the substrate. The chemical characterization of poly (DMA-PFOMA) was tested by a FTIR spectrometer (Nicolet iS10, Thermo Scientific, USA).

### Synthesis of 4-(1-pyrenyl) butyl methacrylate (PBMA)

Ten milliliters dichloromethane, triethylamine (0.17 mL, 1.2 mmol), and 1-pyrenebutanol (0.28 g, 10 mmol) were added to a 50 mL two-neck flask. The flask was kept away from light and the reaction conditions were maintained under nitrogen atmosphere. Methacrylic anhydride (0.18 mL) was slowly added to the reaction solution drop by drop. The solution was stirred for 2 h at 0 °C and then for another 10 h at room temperature to complete the reaction. After the reaction, the solution was extract with HCl (2 mol/L), saturated $NaHCO_3$ and saturated NaCl for three times respectively. Residual water was dried and removed by anhydrous magnesium sulfate. After evaporating the solvent under reduced pressure, the crude product was recrystallized with ethyl acetate, and the final product was successfully synthesized. The chemical structure of PBMA was characterized by $^1H$ NMR (400 MHz) and $^{13}C$ NMR analysis using a Bruker AVANCE III HD NMR spectrometer.

### Preparation of DMCS-hydrogel

The DMCS-hydrogels were synthesized via a low-temperature radical polymerization initiated by ultraviolet light. AAm (3.33, 5.00, and 6.67 wt%, weight of $AAm/H_2O$), AAc (3.50, 7.00, and 10.51 wt%, weight of $AAc/H_2O$), NIPAAm (0, 5.33, 10.66, and 16.00 wt%, weight of $NIPAAm/H_2O$), DMA (0, 0.093, 0.186 wt%, weight of $DMA/H_2O$), Bis (0.0133, 0.0200, 0.0266 wt%, weight of $Bis/H_2O$), 2959 (0.2 wt%, weight of $2959/H_2O$), and were dissolved in 30 g of deionized water to achieve a homogeneous solution. Then, the precursor aqueous solution was poured into a glass mold composed of two glass plates (thickness: 2 mm), and polymerized at 5 °C for 4 h. After removing the template, the DMCS-hydrogel was obtained. Supplementary Table 1 summarizes the codes of the hydrogel and the weight fraction of each component.

### The preparation method of the photo-thermal DMCS-hydrogels

The nanoparticles are doped into the hydrogel prepolymer and polymerized at low temperature. AAm (5.00 wt%, weight of $AAm/H_2O$), AAc (10.51 wt%, weight of $AAc/H_2O$), NIPAAm (16.00 wt%, weight of $NIPAAm/H_2O$), DMA (0.186 wt%, weight of $DMA/H_2O$), Bis (0.0133, weight of $Bis/H_2O$), 2959 (0.2 wt%, weight of $2959/H_2O$), nano-$Fe_3O_4$ (0.03, 0.06, 0.12, 0.24 wt%, weight of nano-$Fe_3O_4/H_2O$) and were dissolved in 30 g of deionized water to achieve a homogeneous solution. Then, the precursor aqueous solution was poured into a glass mold composed of two glass plates (thickness: 2 mm), and polymerized at 5 °C for 4 h.

### Mechanical measurement

The mechanical properties were measured using an electrical universal material testing machine with a 500 N load cell (EZ-Test, Shimadzu). The samples were cut into long strips for testing, and the crosshead velocity was maintained at 100 mm/min. Samples were fully stretched until the material broke, and at the same time a curve of stress versus strain was obtained. The modulus of elasticity (E) was calculated from the stress-strain curve. The elastic modulus was calculated from the slope during 5–15% strain ratio of the stress-strain curve. In order to obtain the stress versus strain curves of the DMCS-hydrogel at high and low temperatures, the hydrogels were tested in hot and cold water. Each experiment was repeated at least three times, and the average value was taken.

### Adhesion performance measurement

The adhesion properties for the hydrogel were measured using an electrical universal material testing machine with a 500 N load cell (EZ-

Test, SHIMADZU) with face-to-face contact mode. The crosshead velocity was kept at 10 mm/min, while the preload was 0.6, 0.8, 1 N and the preload time was 0, 5, 10, 20 s. The probe materials included wood, iron (Fe), glass, silicon wafer (Si), polyethylene glycol terephthalate (PET), aluminum (Al), and titanium (Ti). Each experiment was repeated at least three times, and the average value was taken.

### Friction properties of hydrogel

The 14 FW Statnamic Tribometer (HEIDON Co., Ltd.) friction force testing machine was used to conduct at least three parallel friction tests on each sample. The one-way distance was set to 10 mm, the sliding speed was 10 mm/min, and the applied load was 10 g. The probe was the iron fluorinated with copolymer poly (DMA-PFOMA) (Supplementary Fig. 2). Each experiment was repeated at least three times, and the average value was taken.

### Characterization of Morphologies

The scanning electron microscope (SEM, PhenomPro X, Netherlands) was employed to observe the cross-sectional morphologies of samples. The DMCS-hydrogel sample was cut into a small opening, and then the hydrogel sheet was torn apart in the direction of pre-stretching to obtain a fresh cross section. The preliminarily processed sample was then processed into an anhydrous state by freeze-drying technology. The specific freeze-drying process was to put the torn sample into liquid nitrogen for 10 min, and then quickly transferred to a freeze-dryer and dried in a pressure environment of 1 Pa for 48 h.

### Fluorescence observation

The synthesized fluorescent monomers PBMA (Supplementary Fig. 3) were added to the precursor aqueous solution, and the solution was polymerized under ultraviolet light. The obtained hydrogels contained fluorescence, which was monitored with a confocal laser scanning microscope (Olympus FV1200, Japan).

### Surface morphology characterization

The surface area roughness parameters (Sa) were measured by a 3D VHX-6000 (KEYENCE Corporation, Japan) digital microscope. The digital microscope with panoramic capability at ×20 magnification was used to capture digital images of each test specimen which were placed on the cold (-10 °C) and hot (-40 °C) stage. During the operation of the experiment, an area of 7500 × 7000 µm was completely captured, and with the built-in software the VHX-6000 can stitch the fused images into a large panoramic image. We defined a square measurement zone of 5000 µm of side, which allows us to record the values of Sa (Surface area roughness parameter).

### Statistics and reproducibility

We state that at least three times, each experiment was repeated independently with similar results in our study.

## Data availability

The data supporting the findings of this study are available within the paper, Supplementary Information, and Supplementary Movies. Source data are provided with this paper.

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

## Acknowledgements

We thank our colleagues from the Lanzhou Institute of Chemical Physics, Chinese Academy of Sciences for support and assistance in the research. F.Z., Y.M., and B.Y. acknowledge the funding support from the National Science Foundation of China (22032006, 22102201, and 22072169), National Key Research and Development Program of China (2021YFA0716304), Key Research Project of Shandong Provincial Natural Science Foundation (ZR2021ZD27), and Gansu Province Basic Research Innovation Group Project (22JR5RA093). Y.M. acknowledges support from the Major Program of the Lanzhou Institute of Chemical Physics, CAS (No. ZYFZFX-2) and the Special Research Assistant Project of the Chinese Academy of Sciences.

## Author contributions

Y.M. and F.Z. conceived the idea. Y.M. and Z.Z. designed the experimental protocol. Z.Z. performed and completed the entire experimental studies. C.Q., H.F., Y.X., B.Y., and X.P. provided technical suggestions. Z.Z., Y.M., and F.Z. wrote the paper and all authors discussed the manuscript. Y.M. and F.Z. supervised the entire research.

## Competing interests

The authors declare no competing interests.
