## [Peer Review File · Nature Communications]

Design of large-span stick-slip freely switchable hydrogels via dynamic multiscale contact synergyReviewers' Comments:

Reviewer #1:

Remarks to the Author:

This manuscript describes a switchable adhesive hydrogel via dynamic multiscale contact synergy (DMCS-hydrogel). The DMCS-hydrogel was made by copolymerizing poly(dopamine methacrylamide), poly(N-isopropylacrylamide), poly(acrylamide), and poly(acrylic acid) to make p(AAm-co-AAc-co-NIPAAm-co-DMA). Below the LCST of pNIPAAm, they form intermolecular hydrogen bonding with water molecules, and catechol groups are preferentially exposed, increasing the adhesion at the micro-scale. Meanwhile, above the LCST, as water molecules escape to the surface of the hydrogel, the carboxyl group moves to the surface to shield the adhesive catechol group and increase the lubricating effect, and thus, decreasing the adhesion property. As a result, DMCS-hydrogel shows high adhesion force (21 kPa) at low temperature, and low adhesion force (0.4 kPa) at high temperature with the high switching ratio ~ 240 . They applied DMCS-hydrogel to a climbing robot to realize a fast adhesion switching according to the temperature change. This study exhibits very large on/off adhesion ratio and fast response time comparing to the other research. It can be accepted to "Nature Communications" after some revisions as following.

1. In Fig. 1d, the sample without dopamine methacrylamide (DMA) exhibits adhesion ~ 13 kPa. The authors should explain more clearly why the sample without catechol group has certain amount of adhesion performance.
2. In Fig. 1e, the authors mentioned that they measured the modulus with the same content of catechol group. It is necessary to specify how much DMA was contained in those samples.
3. In Fig. 2, please specify the exact temperature values rather than just high T and low T.
4. In page 5, the author describes as "snapshots of the contact images are obtained at 0, 0.4, 1, and 4s" while Fig. 2b shows images at 0, 0.1, 0.4, and 4 s.
5. In Fig. 2c, high temperature sample also exhibits more than 80% contact area after 20 s. In this case, does the adhesive property at off-state (high T) increase as well? In addition, as shown in Fig. 3b, the authors showed the adhesion strength up to 10 s of contact time. Can you measure the adhesion strength after 20 s?
6. In Fig. 2e, it would be better to add additional image after 10th cycles.
7. In Fig. 2h, hydrophobicity and hydrophilicity were confirmed by the diffusion region, and it seems to be good to add contact angle measurements.
8. In Fig. 3b, the authors should clarify how much preload was applied. Also, in Fig. 3c, the authors should clarify the contact time for the measurements.
9. Please specify the contact time in Fig. 3d. In addition, when performing cyclic tests, is it required to rehydrate the sample to prevent dehydration of the hydrogel?
10. In Supplementary Video 1, 40 °C sample was mislabeled as 'low temperature'.
11. How did you measure the surface roughness of hydrogel in Fig. S6? More detailed explanation about the measurement method is required.

Reviewer #2:

Remarks to the Author:

The manuscript reported the preparation of switchable adhesive hydrogels by copolymerizing several monomers with different functions, and applied them to the track of crawling robot. The design of different components allows the gel can be quickly and reversibly switched between the adhesive state and the non-adhesive state, which largely extends the usable range of reversible adhesion materials.

1. Dopamine groups are easily oxidized in air, and what approach the authors adopted to ensure the stability of the gel over multiple adhesion de adhesion cycles. It is suggested to add the experiment of long-time adhesion stability.
2. Lack of innovation in the design of chemical structure. The chosen and combination of functional monomers (DMA as the adhesive component, NIPAAm as the temperature responsive component, PAAm and PAAc as the elastic and lubricating component) have been reported in previous articles (Nat

Commun 8, 2218 (2017), Adv. Mater. 2018, 30, 1801595), and are well known within the community.

3. It is difficult to control gel adhesion and desorption in real environments by changing the temperature of the external interface. Whether it is possible to change the adhesion state by heating the gel in other forms, such as magnetocaloric or photothermal.

Point-by-Point Response to Review Comments

Dear Reviewers,

We would like to thank you for the constructive and helpful comments and suggestions on the manuscript. We have carefully revised our manuscript and supplementary information according to the comments. Our point-to-point responses are presented below. Changes to the manuscript are shown by red color highlighting. Thank you very much for your help in improving this manuscript.

Please see the “**List of Revision.pdf**” for the full list of revised content in revised manuscript and supplementary information (SI).

In this response letter below,

Response to Reviewer 1.....Page 2-10

Response to Reviewer 2.....Page 10-18

- Authors’ responses are in **blue color**.

-Changes made in revised manuscript and supplementary information are in **red color**.

Thank you for your kind consideration!

Reviewer #1:

This manuscript describes a switchable adhesive hydrogel via dynamic multiscale contact synergy (DMCS-hydrogel). The DMCS-hydrogel was made by copolymerizing poly(dopamine methacrylamide), poly(N-isopropylacrylamide), poly(acrylamide), and poly(acrylic acid) to make p(AAm-co-AAc-co-NIPAAm-co-DMA). Below the LCST of pNIPAAm, they form intermolecular hydrogen bonding with water molecules, and catechol groups are preferentially exposed, increasing the adhesion at the micro-scale. Meanwhile, above the LCST, as water molecules escape to the surface of the hydrogel, the carboxyl group moves to the surface to shield the adhesive catechol group and increase the lubricating effect, and thus, decreasing the adhesion property. As a result, DMCS-hydrogel shows high adhesion force (21 kPa) at low temperature, and low adhesion force (0.4 kPa) at high temperature with the high switching ratio ~240. They applied DMCS-hydrogel to a climbing robot to realize a fast adhesion switching according to the temperature change. This study exhibits very large on/off adhesion ratio and fast response time comparing to the other research. It can be accepted to "Nature Communications" after some revisions as following.

Authors' reply: Many thanks for your positive comments.

Comment 1: *In Fig. 1d, the sample without dopamine methacrylamide (DMA) exhibits adhesion ~13 kPa. The authors should explain more clearly why the sample without catechol group has certain amount of adhesion performance.*

Authors' reply: We thank the reviewer for the professional comment. It is well known that dopamine via the van der Waals forces, hydrogen bonds, coordination or anion/cation- π interactions and so on, can form weak interactions on the surface of substrates to achieve interfacial adhesion. In our work, dopamine methacrylamide (DMA) monomer plays an important role in the adhesion of the DMCS-hydrogel. However, there are also some weak interactions based on other functional groups, such as van der Waals forces originating from the carbon-carbon backbone of polymer which contributes to a certain amount of adhesion without catechol group^{1,2}. What is more, these weak interactions can be annihilated through the presence of surface water molecules at high temperature³.

Moreover, we have made the following modifications in the revised manuscript (Page 4, Line 127): "The adhesion performance of the hydrogels without PDMA may be related to some weak interactions based on other functional groups (e.g. van der Waals forces originating from carbon-carbon backbone)², which are easily annihilated by water molecules³."

Comment 2: In Fig. 1e, the authors mentioned that they measured the modulus with the same content of catechol group. It is necessary to specify how much DMA was contained in those samples.

Authors' reply: We really appreciate the reviewer's suggestion. As mentioned in the manuscript, the modulus of hydrogels is changed only by slightly regulating the crosslinking agent. Among them, these samples have the same DMA content of 0.186 wt% (weight of DMA/H₂O). The specific formula of hydrogel polymerization solution is shown in the following table. Constant DMA content and fine-tuned crosslinker content are highlighted.

Table R1. The weight fraction of DMA in the DMCS-hydrogel.

Sample code	AAc ω_1 (wt%)	AAm ω_2 (wt%)	NIPAAm ω_3 (wt%)	DMA ω_4 (wt%)	Cross-linker ω_5 (wt%)	Initiator Ω_6 (wt%)	Water ω_w (g)
8	10.51	5.00	16.00	0.186	0.0133	0.2	30
11	10.51	5.00	16.00	0.186	0.0200	0.2	30
12	10.51	5.00	16.00	0.186	0.0266	0.2	30

In order to show the content of each monomer in hydrogels included in this work more clearly, we have added a brief description of all sample formulas in the revised Supplementary Information (Page 3, Line 86): **“Formulation of DMCS-hydrogels.** In order to study the effect of each monomer on the performance of hydrogel. We adopted the control variable method to carry out the experiment. The pre-polymer liquid of DMCS-hydrogel is shown in the table below, where the weight percentages of AAc, AAm, NIPAAm, DMA, initiator and cross-linking agent are given with respect to the water solution. ω_w is the water weight.”

Table R2. (Supplementary Table 1 in revised SI) The weight fraction of each component in the DMCS-hydrogel.

Sample code	AAc ω_1 (wt%)	AAm ω_2 (wt%)	NIPAAm ω_3 (wt%)	DMA ω_4 (wt%)	Cross-linker ω_5 (wt%)	Initiator Ω_6 (wt%)	Water ω_w (g)
1	3.50	3.33	10.66	0.186	0.0133	0.2	30
2	7.00	3.33	10.66	0.186	0.0133	0.2	30
3	10.51	3.33	10.66	0.186	0.0133	0.2	30
4	10.51	5.00	10.66	0.186	0.0133	0.2	30
5	10.51	6.77	10.66	0.186	0.0133	0.2	30
6	10.51	5.00	0.00	0.186	0.0133	0.2	30

7	10.51	5.00	5.33	0.186	0.0133	0.2	30
8	10.51	5.00	16.00	0.186	0.0133	0.2	30
9	10.51	5.00	16.00	0.00	0.0133	0.2	30
10	10.51	5.00	16.00	0.093	0.0133	0.2	30
11	10.51	5.00	16.00	0.186	0.0200	0.2	30
12	10.51	5.00	16.00	0.186	0.0266	0.2	30

Comment 3: In Fig. 2, please specify the exact temperature values rather than just high T and low T.

Authors' reply: We thank the reviewer's helpful suggestions. In Fig. 2, the low T is 10 °C and the high T is 40 °C. These values have been added to Fig. 2 in revised manuscript.

Fig. R1 (Fig. 2 in revised manuscript) Dynamic multiscale contact mechanism of DMCS-hydrogel.

Comment 4: In page 5, the author describes as “snapshots of the contact images are obtained at 0, 0.4, 1, and 4s” while Fig. 2b shows images at 0, 0.1, 0.4, and 4 s.

Authors' reply: Thank you very much for pointing out this omission. The contact images and time shown in Fig. 2b are correct. The contact time have been corrected in revised manuscript (Page 5, Line 157): “The snapshots of the contact images show a gradual contact process from 0 s, 0.1 s, 0.4 s, to 4 s, wherein the dark region (dark blue) is in contact with the glass, and bright region (light blue) is the untouched part.”

Comment 5: In Fig. 2c, high temperature sample also exhibits more than 80% contact area after 20 s. In this case, does the adhesive property at off-state (high T) increase as well? In addition, as shown in Fig. 3b, the authors showed the adhesion strength up to 10 s of contact time. Can you measure the adhesion strength after 20 s?

Authors' reply: Thanks for the reviewer's comment. We tested the adhesion strength with a holding time of 20 s. As mentioned by the reviewer, the adhesion strength of the high-temperature samples did slightly increase after 20 s. However, the meso-scale contact did not induce a massive increase in adhesion strength, which might be related to the closed state of the micro-scale contact.

Fig. R2 The relationship between contact time and adhesion. **a** As the contact time increases, the adhesion force increases. **b** Enlarged view of the former.

Comment 6: In Fig. 2e, it would be better to add additional image after 10th cycles.

Authors' reply: Thank you for your helpful suggestion. As suggested by the reviewers, we increased the number of low and high temperature cycles to 50 times. The reversible cycling performance of DMCS-hydrogel does not weakened after 50 cycles. Among them, in order to prevent the hydrogel from losing water, the hydrogel was placed in water for 5 s to restore each 10 cycles, and then kept at 10 °C temperature and 60% humidity for 6 h. Moreover, relevant experimental information has been added to the revised Supplementary Information (Page 11, Line 239): “The reversible cycling performance of DMCS-hydrogel does not weakened after 50 cycles. Among them, in order to prevent the hydrogel from losing water, the hydrogel was

placed in water for 5 s to restore each 10 cycles, and then kept at 10 °C temperature and 60% humidity for 6 h.”

Fig. R3 (Supplementary Fig. 13 in revised Supplementary Information). **Confocal laser scanning microscope images under low and high temperature conditions.** At low temperature, the hydrogel showed strong fluorescence, but at high temperature, the fluorescence disappeared. Scale bar 100 μ m.

Fig. R4 (Supplementary Fig. 14 in revised Supplementary Information). The Mean Integrated Optical Density of hydrogels at high and low temperature conditions.

Comment 7: *In Fig. 2h, hydrophobicity and hydrophilicity were confirmed by the diffusion region, and it seems to be good to add contact angle measurements.*

Authors’ reply: We thank the reviewer for the great suggestion. The contact angles of DMCS-hydrogel are tested under high and low-temperature conditions. The results showed that the high-temperature contact angle is smaller than the low-temperature. Moreover, the corresponding contact angle decreases faster on the high-temperature hydrogel than on the low-temperature hydrogel. All results are consistent with the diffusion law of droplets at the interface.

Moreover, in the revised manuscript (Page 6, Line 193), we have also added this

part of data to the manuscript. The specific description is as follows: “The process of dynamic wetting in the vertical and horizontal directions was recorded using microscopes, with the wetting radius r_I at low temperature and z_I at high temperature. As shown in Fig. 2h and Supplementary Video 2, the water droplets spread faster on the high-temperature hydrogel than on the low-temperature hydrogel from the top view and the corresponding contact angles of high temperature decrease faster than that of the low temperature from the side view (Supplementary Fig. 15 and Fig. 16). Specifically, the diffusion rate at high temperature ($23.4 \times 10^3 \mu\text{m}^2/\text{s}$) was 3 times higher than that at low temperature ($7.2 \times 10^3 \mu\text{m}^2/\text{s}$), indicating that the high-temperature surface has more hydrophilic functional groups (Fig. 2i).”

Fig. R5 (Supplementary Fig. 15 in revised Supplementary Information). **Dynamic wetting process under high and low temperature conditions.** Water droplet were dripped on the high and low-temperature hydrogels to test their dynamic wettability.

Fig. R6 (Supplementary Fig. 16 in revised Supplementary Information). **Relationship between contact angle and time.** The contact angle is smaller at high temperature than at high temperature. Water droplet spreads faster on the high-temperature DMCS-hydrogel than on the low-temperature.

Comment 8: In Fig. 3b, the authors should clarify how much preload was applied. Also, in Fig. 3c, the authors should clarify the contact time for the measurements.

Please specify the contact time in Fig. 3d.

Authors' reply: We thank the reviewer's helpful suggestions. In Fig. 3b, the preload was 10 kPa. In Fig. 3c, the contact time was 10 s. Based on the above conditions, we explored the influence of contact load and contact time on adhesion strength. Unless otherwise specified, all adhesion test conditions are 10 s contact time and 10 kPa preload in this work.

To help readers know the test condition more clearly, we have added a brief description in the revised manuscript (page 7, Line 204) as "To further investigate the effect of temperature, the adhesion properties of hydrogel to glass was detected at different temperatures from 10 °C to 80 °C (Fig. 3a, Supplementary Fig. 17 and Fig. 18). Unless otherwise specified, all adhesion test conditions are 10 s contact time and 10 kPa preload." and in the revised manuscript (page 7, Line 214) as "For this multiscale adhesion system, the adhesion strength of DMCS-hydrogel increased slightly with the increase of contact time (constant preload: 10 kPa) and preload (constant contact time: 10 s) at low temperature, and exhibited negligible change at high temperature, indicating that the rearrangement of the molecular conformation on the hydrogel surface and interfacial contact were completed in a very short time, which resulted in the fast switching of the adhesion with temperature alternation (Fig. 3b, 3c and Supplementary Fig. 29). Unless otherwise specified, all adhesion test conditions are 10 s contact time and 10 kPa preload." Moreover, we marked the preload and contact time on Figures 3b and 3c. As shown below:

Fig. R7 (Fig. 3 in revised manuscript). **a** The relationship between the temperature and adhesion strength of DMCS-hydrogel. The effects of contact time **b** and preload **c** on adhesion strength at high and low temperatures against a glass probe.

Comment 9: In addition, when performing cyclic tests, is it required to rehydrate the sample to prevent dehydration of the hydrogel?

Authors' reply: We thank the reviewer for the insightful and professional comment. In Fig. 3d, when performing cyclic tests, samples do not need to be rehydrated.

However, as the reviewer commented, the problem of water loss from hydrogels is a major challenge in hydrogel applications⁴. Therefore, to test the limits of our material, we performed a more number of adhesion cycles. The results showed that after 15 times, both high and low adhesion properties of the hydrogels were gradually increased, especially low-temperature adhesion, which may be caused by the water loss of the hydrogels (Fig. R8). Interestingly, this increase in the adhesion performance can be restored through a simple rehydration method. After 10 cycles of adhesion, the hydrogel was placed in water for 5 s to restore, and then kept at 10 °C temperature and 60% humidity for 6 h, and its adhesion performance will be restored to its original state. This continued for 50 cycles, and its adhesion performance remained stable (Fig. R9).

Moreover, in the revised manuscript (Page 7, line 227), we have also added this part of data to the manuscript. The specific description is as follows: “Under a condition of simple rehydration to prevent water loss, the adhesion performance of DMCS-hydrogel can be stably and reversibly switched more than 50 times (Supplementary Fig. 20).”

Fig. R8 (Supplementary Fig. 19 in revised Supplementary Information). The switchable adhesion cycles of DMCS-hydrogel at high (60 °C) and low (10 °C) temperature for 15 cycles.

Fig. R9 (Supplementary Fig. 20 in revised Supplementary Information). The switchable adhesion cycles of DMCS-hydrogel at high (60 °C) and low (10 °C) temperature for 50 cycles (in water for 5 s, at 10 °C temperature and 60% humidity for 6 h, every 10 cycles).

Comment 10: *In Supplementary Video 1, 40 °C sample was mislabeled as ‘low temperature’.*

Authors’ reply: Thanks to reviewer for pointing out this careless mistake. Thank you for your kind reminder. Supplementary video 1 has been modified.

Fig. R10. The screenshot of revised Supplementary video 1 (the contact evolution of DMCS-hydrogel).

Comment 11: *How did you measure the surface roughness of hydrogel in Fig. S6? More detailed explanation about the measurement method is required.*

Authors’ reply: We thank the reviewer for this necessary comment. The surface roughness of hydrogels in the manuscript and supplementary information are measured by 3D VHX-6000 (KEYENCE Corporation, Japan) digital microscope, which can capture the 3D surface morphology of the hydrogels.

Moreover, the more detailed explanation about the roughness measurement method has been supplemented in the revised manuscript (Page 12, Line 387). The specific description is as follows: “**Surface morphology characterization.** The surface area roughness parameters (Sa) were measured by a 3D VHX-6000 (KEYENCE Corporation, Japan) digital microscope. The digital microscope with 3D panoramic capability at $\times 20$ magnification was used to capture digital images of each test specimen which were placed on the cold (~ 10 °C) and hot (~ 40 °C) stage. During the operation of the experiment, an area of 7500×7000 μm was completely captured, and with the built-in software the VHX-6000 can stitch the fused images into a large panoramic image. We defined a square measurement zone of 5000 μm of side to record the values of Sa (Surface area roughness parameter).”

Reviewer #2:

The manuscript reported the preparation of switchable adhesive hydrogels by copolymerizing several monomers with different functions, and applied them to the track of crawling robot. The design of different components allows the gel can be

quickly and reversibly switched between the adhesive state and the non-adhesive state, which largely extends the usable range of reversible adhesion materials.

Authors' reply: Thank you very much for your helpful comments.

Comment 1: *Dopamine groups are easily oxidized in air, and what approach the authors adopted to ensure the stability of the gel over multiple adhesion de adhesion cycles. It is suggested to add the experiment of long-time adhesion stability.*

Authors' reply: Thank you for your constructive and helpful suggestion. It is well known that *Dopamine* (catechol groups) is a very critical and important adhesion functional group in mussel adhesion⁵. In this work, the mussel-inspired adhesion component poly (dopamine methacrylamide) (PDMA) with catechol groups was applied to the DMCS-hydrogel via radical polymerization, which has an important contribution to the surface adhesion of the hydrogel. As the reviewer commented, dopamine and its derivatives can be oxidized in air, but the rate of this oxidation is slow. To demonstrate this, we design three experiments to study the oxidation of PDMA in DMCS-hydrogels: (i) Transmission, (ii) FT-IR spectroscopy and (iii) Adhesion performance verification.

(i) Transmission verification: It has been found that the color of the polymer containing catechol groups will deepen once the catechol groups are oxidized to quinone. Therefore, the molecules containing catechol groups such as dopamine or its derivatives can be used for hair dyeing by means of oxidation⁶. Based on this, the transmittance of the DMCS-hydrogels were used to detect its degree of oxidation. The hydrogels with PDMA and without PDMA were placed in air for 120 h (5d) (Fig. R11a and R11c). The hydrogels oxidized by H₂O₂ solution were used as control samples (Fig. R11b and R11d). The results show that the transmittance of DMCS-hydrogels containing PDMA decreases dramatically after oxidation in H₂O₂ solution for 12 h. In contrast, the transmittance of hydrogels without PDMA in the oxidation solution hardly fluctuates. Similarly, the hydrogel placed in air still maintains its transparent color, and the transmittance is almost unchanged even after 120 h (5d). Therefore, we speculate that the oxidation of PDMA in DMCS hydrogels can be negligible after 5 days in air.

Fig. R11. UV transmittance of hydrogels (with and without DMA) in air and H₂O₂ solution. **a** The transmittance of the hydrogel containing PDMA in air. **b** The transmittance of the hydrogel containing PDMA in H₂O₂ solution. **c** The transmittance of the hydrogel without PDMA in air. **d** The transmittance of the hydrogel without PDMA in H₂O₂ solution.

(ii) Infrared spectrum verification: The hydrogels containing PDMA were placed in air and H₂O₂ solution respectively, and the evolution of the characteristic peaks of each group was recorded. As a result, for DMCS-hydrogels in H₂O₂ solution, with regard to the broad peak appearing at 3200-3500 cm⁻¹ due to O-H vibration, the peak intensity gradually decreases and the characteristic peaks moved to higher wave numbers with increasing soaking time. The reason is the reduction of the number of hydroxyl groups on the phenyl ring and the conversion of -OH···O-C- catechol hydrogen bond to -OH···O=C- quinone hydrogen bonding^{7, 8}. Moreover, the peak at 1170 cm⁻¹ by the C-O vibration evidently decreased with increasing soaking time.^{7, 8} All the above results show the formation of quinone, indicating that PDMA can be oxidized in H₂O₂ solution. However, for DMCS-hydrogels in air, the characteristic peak of the infrared spectrum hardly fluctuates and changes even after 120 h (5d). The above experiments demonstrated that PDMA in DMCS-hydrogels is hardly oxidized after 5 days in air (Fig. R12).

Fig. R12. Infrared spectrum of hydrogels (with PDMA). DMCS-hydrogels are placed in air and H₂O₂ solution.

Adhesion verification: To further verify the long-time adhesion stability, we tested the adhesion performance of DMSC-hydrogel for 5 days. As expected, the adhesion performance of the hydrogel remained stable during the 5-day adhesion test without a significant decline (Fig. R13). This also signifies that the PDMA contained in the DMCS-hydrogel is hardly oxidized during prolonged exposure to air. In order to reduce the error caused by the water loss of the sample, the samples were stored in a constant temperature and humidity chamber (10 °C temperature and 60 % humidity) after each test.

Moreover, in the revised manuscript (Page 7, line 227), we have also added this part of data to the manuscript. The specific description is as follows: “Under a condition of simple rehydration to prevent water loss, the adhesion performance of DMCS-hydrogel remained stable during the 5-day adhesion test without a significant decline (Supplementary Fig. 21).”

Fig. R13 (Supplementary Fig. 21 in revised Supplementary Information). Long-term adhesion stability of DMSC-hydrogels (10 °C temperature and 60 % humidity).

Comment 2: *Lack of innovation in the design of chemical structure. The chosen and combination of functional monomers (DMA as the adhesive component, NIPAAm as the temperature responsive component, PAAm and PAAc as the elastic and lubricating component) have been reported in previous articles (Nat Commun 8, 2218 (2017), Adv. Mater. 2018, 30, 1801595), and are well known within the community.*

Authors' reply: Thank you very much for your comments. In this work “Design of large-span stick-slip freely switchable hydrogels via dynamic multiscale contact synergy”, there are essential differences from the previous two works, including (i) mechanisms, (ii) adhesion performance, and (iii) functional monomers (chemical structure).

(i) Innovation of mechanisms: Dynamic multiscale contact synergy: new conceptual design with *dynamic* multiscale contact synergy (this work) vs. dynamic microscale contact⁹ (Nat. Commun., 2017) or dynamic microscale contact + non-dynamic microstructure¹⁰ (Adv. Mater., 2018).

Table R3. Mechanism comparison of three works

Work	Key Innovations	Mechanism
Nat. Commun. (2017)	reversible temperature-responsive adhesion regulation	dynamic microscale contact
Adv. Mater. (2018)	remote control over surface adhesion	dynamic microscale contact (non-dynamic microstructure)
This work	large-span stick-slip reversible switching via dynamic multiscale contact synergy	dynamic microscale + dynamic mesoscale contact synergy

(ii) New record of smart adhesion:

For DMCS-hydrogels in this work, the dynamic multiscale contact synergy endows the hydrogels with (i) large-span adhesion regulation; (ii) switchable and dynamic adhesiveness between the slippery (friction ~ 0.04 N/cm²) and the sticky (adhesion ~ 3 N/cm²) states for solid-solid contact; (iii) fast attachment and detachment without residue (<1s).

Fig. R14. Comparison of adhesion properties between DMCS-hydrogel and two smart adhesive materials reported in Nat. Commun. (2017) and Adv. Mater. (2018).

(iii) Differences of functional monomers:

As shown in Table R4, the adhesion materials reported in these three works are composed of different functional monomers. Especially, compared with the two previous works, we introduced lubricating functional component [poly (acrylic acid) (AAc)] and elastic functional component [poly(acrylamide) (PAAm)] for the first time and achieved stick-slip switching in this work. Moreover, the novelty of this work is to propose a dynamic multiscale contact coupling concept, which provides a new framework for smart adhesion hydrogel with large-span, switchable, dynamic, and fast adhesion regulation [Yet there has not been a hydrogel material that can simultaneously regulate the contact at each scale to achieve a "resonant-like" multi-scale contact coupling based on a same variate. The present work truly brings the synthetic materials with synergy of dynamic multiscale contact, which has been called for by the community on 2018 (Nature 559, 77-82 (2018).)]. What is more, the contact mechanism of soft matter is explored and revealed in this work. The findings with experimental proof advanced the understanding and new mechanism of achieving excellent adhesion performance based on contact for polymeric soft materials.

Table R4. Functional monomers used in three works

Work	Functional monomers
Nat. Commun. (2017)	dopamine methacrylamide (DMA) ; adamantine (AD) ; methoxyethyl acrylate(MEA); amino-β-cyclodextrin (CD); N-isopropyl acrylamide (NIPAAm)
Adv. Mater. (2018)	methoxyethyl acrylate (MEA); dopamine methacrylamide (DMA); N-isopropyl acrylamide (NIPAAm)
This work	acrylamide (AAm); acrylic acid (AAc); dopamine methacrylamide (DMA); and N-isopropyl acrylamide (NIPAAm)

Please see the detailed description about the previous work in the original manuscript (page 2, line 52 to 61) as: “For the design of switchable adhesive materials at the micro-scale, a representative example involves mussel-inspired biomimetic method, which is normally based on reversible screening of the catechol groups at the molecular level [Nat. Commun. (2017)]. However, the change of contact area at the molecular level cannot be guaranteed without contact at other scales, thus uncontrollable/chaotic contacts at other scales may weaken the switching of adhesiveness, leaving a substantial portion of multiscale contact underutilized in contact switching during adhesion regulation. Incorporating mussel-mimetic smart polymers into micro/nano-structures could increase the effective contact area at the mesoscopic scale and improve high-low adhesion strength at the same time. Unfortunately, structure-assisted smart adhesives essentially only rely on adhesion regulation at the molecular scale [Adv. Mater. (2018)].”

Comment 3: *It is difficult to control gel adhesion and desorption in real environments by changing the temperature of the external interface. Whether it is possible to change the adhesion state by heating the gel in other forms, such as magnetocaloric or photothermal.*

Authors' reply: We thank the reviewer for the insightful comment and helpful suggestion. In our experiments, adhesion properties of DMCS-hydrogels are reversibly switched by temperature. As suggested by the reviewer, theoretically, any method that can control the temperature, such as magnetocaloric, electrocaloric, photothermal, can be utilized to achieve reversible switching of adhesion properties. Here, we tried to use Fe₃O₄ nanoparticles (NPs) doped in the hydrogels to achieve adhesion regulation based on photothermal effect.

As shown in Fig. R15a, the temperature of Fe₃O₄ nanoparticles-containing DMCS-hydrogels increases rapidly during near-infrared light for 2 min irradiation. Moreover, the temperature increased faster with the increase of the Fe₃O₄ nanoparticles content (Fig. R16 and Fig. R17). In contrast, the temperature of the hydrogels without nanoparticles showed hardly any increase under light irradiation (Fig. R15b). What is more, the doping of nanoparticles in the hydrogel did not affect its adhesion properties. As expected, the adhesion properties of the hydrogel with nano-Fe₃O₄ can be reversibly switched by light irradiation (Fig. R18).

Moreover, in the revised manuscript (Page 8, Line 253), we had also discussed about other control methods for adhesion performance of the hydrogels, as follows: **“Moreover, we incorporated photo-thermal Fe₃O₄ nanoparticles in the hydrogel. The adhesion properties of hydrogels were managed by utilizing local and remotely**

controlled temperature responses (Supplementary Fig. 26).”

In the revised supplementary information (Page 15, Line 307), the photothermal effect of the DMCS-hydrogels (with Fe₃O₄) were added, as follows: “The photothermal effect of the DMCS-hydrogels (with Fe₃O₄): The adhesion properties of hydrogel with nano-Fe₃O₄ can be reversibly switched by light irradiation (Supplementary Fig. 26). As shown in Supplementary Fig. 25, the temperature of the nanoparticles-containing hydrogels increases rapidly during near-infrared light for 2 min irradiation. Moreover, the temperature increased faster with the increase of the nanoparticles content (Supplementary Fig. 24 and Fig. 25). In contrast, the surface temperature of the hydrogels without nanoparticles showed hardly any increase under light irradiation.”

In the revised manuscript (Page 11, Line 345), the specific preparation method of the DMCS-hydrogels (with Fe₃O₄) were added, as follows: “**The preparation method of the photothermal DMCS-hydrogels:** The nanoparticles are doped into the hydrogel prepolymer and polymerized at low temperature. AAm (5.00 wt%, weight of AAm/H₂O), AAc (10.51 wt%, weight of AAc/H₂O), NIPAAm (16.00 wt%, weight of NIPAAm/H₂O), DMA (0.186 wt%, weight of DMA/H₂O), Bis (0.0133, weight of Bis/H₂O), 2959 (0.2 wt%, weight of 2959/H₂O), nano-Fe₃O₄ (0.03, 0.06, 0.12, 0.24 wt%, weight of nano-Fe₃O₄/H₂O) and were dissolved in 30 g of deionized water to achieve a homogeneous solution. Then, the precursor aqueous solution was poured into a glass mold composed of two glass plates (thickness: 2 mm), and polymerized at 5 °C for 4 h.”

Fig. R15 (Supplementary Fig. 23 in revised Supplementary Information). The DMCS-hydrogels with 0.24 wt% nanoparticles (a) were gradually heated based on photothermal effect. The hydrogels without NPs were used as control samples.

Fig. R16 (Supplementary Fig. 24 in revised Supplementary Information). The relationship between the temperature of hydrogels and nanoparticle content under 2 min irradiation.

Fig. R17 (Supplementary Fig. 25 in revised Supplementary Information). **a.** Relationship between temperature and nanoparticle content after 2 min light irradiation. **b.** The relationship between the temperature of the DMCS-hydrogels with 0.24 wt% NPs and the irradiation time. The hydrogels without NPs were used as control samples.

Fig. R18 (Supplementary Fig. 26 in revised Supplementary Information). Adhesion properties of DMCS-hydrogels (with nano-Fe₃O₄) under light irradiation and without light irradiation. TEM image shows the morphology of nanoparticles.

Reference

1. Bowden, P. B. The elastic modulus of an amorphous glassy polymer. *Polymer* **9**, 449-454 (1968).
2. Wang, K., Pang, X. & Cui, S. Inherent stretching elasticity of a single polymer chain with a carbon-carbon backbone. *Langmuir* **29**, 4315-4319 (2013).
3. Kendall, K. & Roberts, A. D. van der Waals forces influencing adhesion of cells. *Philos. Trans. R. Soc. B. Biol. Sci.* **370**, 20140078 (2015).
4. Li, Z. et al. Gelatin methacryloyl-based tactile sensors for medical wearables. *Adv. Funct. Mater.* **30**, 2003601 (2020).
5. Lee, B. P., Messersmith, P. B., Israelachvili, J. N. & Waite, J. H. Mussel-Inspired Adhesives and Coatings. *Annual review of materials research* **41**, 99-132 (2011).
6. Morel, O. J & Christie, R. M. Current trends in the chemistry of permanent hair dyeing. *Chem. Rev.* **111**, 2537-2561 (2011).
7. Xia, J., Xu, Y., Lin, J. & Hu, B. UV-induced polymerization of urushiol without photoinitiator. *Progress in Organic Coatings* **61**, 7-10 (2008).
8. Shin, M. et al. Complete prevention of blood loss with self-sealing haemostatic needles. *Nat. Mater.* **16**, 147-152 (2017).
9. Zhao, Y. et al. Bio-inspired reversible underwater adhesive. *Nat. Commun.* **8**, 1-8 (2017).
10. Ma, Y. et al. Remote Control over Underwater Dynamic Attachment/Detachment and Locomotion. *Adv. Mater.* **30**, e1801595 (2018).

Reviewers' Comments:

Reviewer #1:

Remarks to the Author:

Authors addressed all the comments raised by the reviewers.

Reviewer #2:

Remarks to the Author:

It has been well revised, I suggest accept of the manuscript

Reply to the Reviewers

Reviewer #1:

Comment 1: Authors addressed all the comments raised by the reviewers.

Authors' reply: We thank the reviewer's kind recognition of our work and comments.

Reviewer #2

Comment 1: It has been well revised, I suggest accept of the manuscript

Authors' reply: We thank the reviewer's kind recognition of our work and comments.